

# The effect of cloud liquid water on tropospheric temperature retrievals from microwave measurements

Leonie Bernet [1,2], Francisco Navas-Guzmán [1], and Niklaus Kämpfer [1,2]

[1]Institute of Applied Physics, University of Bern, Bern, Switzerland
[2]Oeschger Centre for Climate Change Research, University of Bern, Bern, Switzerland

*Correspondence to:* Leonie Bernet (leonie.bernet@iap.unibe.ch)

**Abstract.** Microwave radiometry is a suitable technique to measure atmospheric temperature profiles with high temporal resolution during clear sky and cloudy conditions. In this study, we included cloud models in the inversion algorithm of the microwave radiometer TEMPERA (TEMPErature RAdiometer) to determine the effect of cloud liquid water on the temperature retrievals. The cloud models were built based on measurements of cloud base altitude and integrated liquid water (ILW), all performed at the aerological station (MeteoSwiss) in Payerne (Switzerland). Cloud base altitudes were detected using ceilometer measurements while the ILW was measured by a HATPRO (Humidity And Temperature PROfiler) radiometer. To assess the quality of the TEMPERA retrieval when clouds were considered, the resulting temperature profiles were compared to two years of radiosonde measurements. The TEMPERA instrument measures radiation at 12 channels in the frequency range from 51 to 57 GHz, corresponding to the left wing of the oxygen emission line complex. When the full spectral information with all the 12 frequency channels was used, we found a marked improvement in the temperature retrievals after including a cloud model. The chosen cloud model influenced the resulting temperature profile, especially for high clouds and clouds with a large amount of liquid water. Using all 12 channels however presented large deviations between different cases, suggesting that additional uncertainties exist in the lower, more transparent channels. Using less spectral information with the higher, more opaque channels only also improved the temperature profiles when clouds where included, but the influence of the chosen cloud model was less important. We conclude that tropospheric temperature profiles can be optimized by considering clouds in the microwave retrieval, and that the choice of the cloud model has a direct impact on the resulting temperature profile.

## 1 Introduction

Measurements of tropospheric temperature with a high temporal resolution are fundamental for climate and weather research, to investigate atmospheric changes and to study dynamic or radiative processes in the atmosphere. Highly resolved temperature information is further needed for weather forecasting and nowcasting, numerical weather prediction models, as well as climate models. A well established in situ technique to measure temperature profiles with high vertical resolution are radiosondes. However, radiosonde measurements have some disadvantages such as limited temporal resolution, high costs and logistical difficulties. Ground-based microwave radiometry has thus been discussed to be a suitable technique to provide continuous temperature measurements with a high temporal resolution (Askne and Westwater, 1986; Solheim et al., 1998; Crewell et al.,





2001). Microwave radiometry has the advantage of being able to measure temperature during clear sky and cloudy weather conditions in contrast to other ground-based remote sensing techniques such as lidars or infrared spectrometers. This is due to the semitransparent properties of clouds in the microwave spectra (Löhnert et al., 2004). However, liquid water in clouds still absorbs microwave radiation at some frequencies (Rosenkranz et al., 1972) and should thus be considered in the temperature
retrievals.

Temperature profiling using microwave radiometers is a well established technique and the performance for tropospheric retrievals has been evaluated in various studies (e.g. Stähli et al. (2013); Löhnert and Maier (2012); Sánchez et al. (2013); Navas-Guzmán et al. (2016)). Nevertheless, the specific effect of cloud liquid water on microwave temperature measurements has only been assessed in few studies (Decker et al., 1978; Cimini et al., 2011; Chan and Lee, 2015). Decker et al. (1978)
attempted a correction of temperature retrievals from a basic three-channel microwave radiometer by considering the absorption of cloud liquid water based on radiosonde measurements. They stated that the correction for clouds can effectively reduce the error of temperature measurements. In more recent times, Cimini et al. (2011) used a microwave radiometer with a new retrieval technique based on outputs from a numerical weather prediction model, also including information on cloud liquid water. However, they did not analyse explicitly the retrieval accuracy for cloudy situations, due to a limited study period. Chan
and Lee (2015) included absorption and emission of cloud liquid water in the temperature retrieval of a microwave radiometer, but their study focusses on the retrieval improvement when rain absorption and scattering is incorporated in the algorithm and the effect of cloud liquid water is not explicitly investigated. The direct effect of cloud liquid water on temperature retrievals from microwave radiometers has thus to be analysed and the improvement of the retrievals when clouds are considered in the algorithm has to be assessed.

A first step towards these objectives has been performed by Navas-Guzmán et al. (2014), who developed an integrated approach to incorporate clouds in temperature retrievals by using data from different cloud instruments with high temporal resolution. By comparing temperature measurements from a microwave radiometer with radiosondes they have shown that the temperature profiles generally improve when a cloud model is considered in the retrieval. However, in their study the radiosonde launches used for verification and the microwave measurements were not performed at the same location. Furthermore, only
a simple cloud model has been tested and the effect of various cloud models different in shape and amount of liquid water has not yet been determined. The present study extends the work from Navas-Guzmán et al. (2014) by further investigating the influence of liquid water on temperature retrievals and by comparing tropospheric temperature profiles with radiosonde measurements at the same location. Our objective is to improve temperature retrievals during cloudy conditions and to examine whether including different cloud models for situations with different cloud properties optimizes the temperature retrieval.

The temperature measurements were performed with TEMPERA (TEMPErature RAdiometer), which is a ground-based microwave radiometer that uses oxygen emission in the atmosphere to retrieve tropospheric and stratospheric temperature profiles (Stähli et al., 2013). The present study focusses on tropospheric retrievals (0-10 km), and stratospheric retrievals are only briefly investigated in Sect. 6. TEMPERA measures rotational transitions of molecular oxygen at 12 frequency channels at the left wing of the 60 GHz oxygen emission feature. Because of the sensitivity of the lower frequencies to liquid water, Stähli
et al. (2013) only used the 8 higher frequency channels when clouds were present. To allow the use of all frequencies with



a larger amount of spectral information also for cloudy conditions, liquid water has to be considered in the retrieval. For this purpose, clouds have been characterized in the present study by using measurements from different instruments available at the regional Centre of the Swiss Federal Office for Meteorology and Climatology (MeteoSwiss) in Payerne (Switzerland) where the microwave radiometer TEMPERA is located since December 2013. We then used the measured cloud properties to include

a cloud model in the tropospheric temperature retrievals of TEMPERA. In order to assess the cloud effect on the retrievals, the resulting temperature profiles were evaluated against retrievals without considering clouds. To determine the performance of the retrievals the results were compared with the temperature profiles from radiosondes which are launched twice a day at the meteorological station in Payerne.

## 2   Instrumentation and data

The microwave radiometer used for temperature measurements in this study as well as the different instruments used for cloud characterization are presented in the following. All instruments are located at the station of MeteoSwiss in Payerne (46.8° N, 7.0° E) in western Switzerland at an altitude of 491 m above sea level (a.s.l.).

### 2.1   TEMPERA

The microwave radiometer TEMPERA retrieves temperature profiles from the ground up to 50 km. It has been designed and
built by the Institute of Applied Physics at the University of Bern (Switzerland) and is the first ground-based microwave radiometer that can retrieve temperature profiles for the troposphere and the stratosphere at the same time (Stähli et al., 2013). The TEMPERA instrument measures radiation emitted by the atmosphere in the frequency range from 51 to 57 GHz. It operates in an isolated, temperature-stabilized room. The radiation coming from the atmosphere penetrates a blue styrofoam window (transparent to microwave radiation) and is reflected by the instrument's mirror into the antenna and the receiver (Fig. 1).

The spectral analysis for the tropospheric retrieval is performed in a filterbank that allows the analysis of 12 different frequency channels. The instrument measures the 12 channels at 9 different zenith angles, from 30° to 70° with 5° angular steps. Table 1 displays the central frequencies used with the corresponding bandwidths. The lower 9 channels (from 51.25 to 55.40 GHz) have a bandwidth of 250 MHz, whereas the bandwidth is 1 GHz for the three more opaque channels (from 56.00 to 57.00 GHz). The larger bandwidth for the higher channels guarantees a higher sensitivity, which is necessary because of the

small spectral dependency for those frequencies close to the center of the 60 GHz oxygen emission peak. For the stratospheric temperature retrievals, a Fast Fourier Transform (FFT) spectrometer is used with a bandwidth of 960 MHz and a spectral resolution of 30.5 kHz. It measures two oxygen emission lines centred at 52.5424 and 53.0669 GHz. A measurement cycle of the instrument lasts 60 s, including a hot load calibration and the atmospheric measurements, first at 30° for the stratospheric retrieval and then at the 9 mentioned zenith angles for the tropospheric retrieval. A mean of 15 measurement cycles for the

tropospheric and of 120 cycles for the stratospheric retrieval is used. The temporal resolution of the retrieval is thus 15 minutes for the troposphere and 2 hours for the stratosphere. Further information about the instrument design and the measurement technique can be found in Stähli et al. (2013).





## 2.2 Radiosondes

Radiosonde measurements are used in this study for validation of the temperature profiles from TEMPERA. The radiosondes (type SRS-C34) are launched twice a day at 11:00 and 23:00 UTC (universal time coordinated) at the MeteoSwiss station in Payerne. The launch usually starts one hour before the official time (12:00 and 00:00 UTC) to cope with the ascent time, that usually lasts two hours until the balloon bursts at around 30 km (Löhnert and Maier, 2012). The troposphere is crossed within approximately 30 minutes, assuming an average ascent rate of $5\,\mathrm{m\,s^{-1}}$.

## 2.3 Instruments for cloud detection

The inclusion of liquid water absorption in the TEMPERA retrieval is based on cloud measurements from different instruments. Taking advantage of the infrastructure available at the MeteoSwiss station in Payerne, the following instruments have been used to obtain cloud information: a lidar ceilometer for cloud base height (CBH), a HATPRO (Humidity And Temperature PROfiler) microwave radiometer for integrated liquid water (ILW), and a hemispherical sky camera for cloud cover and cloud type. Additionally, data from an automatic partial cloud amount detection algorithm (APCADA) was used for information about cloud cover (Dürr and Philipona, 2004). All those instruments are located at the same location as TEMPERA, including the radiosoundings, which is a big advantage for the analysis of highly varying properties such as tropospheric temperature and clouds.

### 2.3.1 Ceilometer

The ceilometer that was used in this study to detect the CBH was a CBME80 ceilometer manufactured by Eliasson. The system is based on a lidar (light detection and ranging) principle, measuring backscattered radiation from an emitted laser pulse in the infrared part of the electromagnetic spectrum. Only the lowest cloud layer detected by the ceilometer is used in this study and multiple cloud layers are thus not considered.

### 2.3.2 HATPRO

The HATPRO dual profiler is a microwave radiometer from Radiometer Physics GmbH (RPG, Germany) that retrieves atmospheric integrated liquid water (ILW) and integrated water vapour (IWV), as well as temperature profiles. It measures radiation at two bands from 22 to 31 GHz (for water) and from 51 to 58 GHz (for temperature). For this study we are only interested in ILW. Besides the ILW, HATPRO also provides a rain detection that was used in this study to exclude rainy cases from the analyses.

### 2.3.3 Sky camera

To determine cloud cover and to obtain cloud type estimations, we used data from a sky camera (CMS Schreder GmbH). The camera system consists of a commercial camera with a fisheye lens that takes pictures from the sky. The detection of clouds by the sky camera is based on algorithms that use red-green-blue (RGB) images, with a threshold to distinguish cloud and





cloud-free pixels. The amount of sky camera data is much smaller than for ceilometer or HATPRO data because sky camera data is only available during day. For more information about the sky camera, please refer to Wacker et al. (2015). Besides the detection of clouds, the sky camera data also provides a cloud classification, based on an algorithm adopted from Heinle et al. (2010). The following seven cloud types are distinguished by the algorithm: cumulus (Cu), cirrus and cirrostratus (Cr-Cs),

cirrocumulus and altocumulus (Cc-Ac), stratocumulus (Sc), stratus and altostratus (St-As), cumulonimbus and nimbostratus (Cb-Ns), and clear sky. In this study, the cloud cover and cloud type have been averaged over 10 minutes. In some cases two different cloud types have been detected in this time span, resulting in various combinations of different cloud types. To reduce the number of different cloud types, we have merged the occurring types to overall 11 groups of cloud types (described in detail in Fig. 5).

### 2.3.4   APCADA

The automatic partial cloud amount detection algorithm (APCADA) is a method to determine cloud cover based on accurate measurements of longwave downward radiation (LDR), temperature, and relative humidity at the surface. The algorithm was developed by Dürr and Philipona (2004) using long time series of LDR measurements at different stations worldwide. An Eppley Precision Infrared Radiometer pyrgeometer is used to determine the LDR, which provides in combination with the

measured surface temperature information about the sky emittance and cloud cover (Dürr and Philipona, 2004; Wacker et al., 2015).

### 2.4   Data set description

Two years of data has been investigated in this study (2014–2015). For the cloud analysis (Sect. 4), we used 10-minutes-means from data of all cloud instruments for the time period covered by this study. Depending on the instrument availability, we

considered over 35000 10-minutes values (number of cases denoted by $n$) for some instruments.

To include cloud models in the TEMPERA retrieval (Sect. 5), the cloud data has been averaged over 15 minutes to correspond to the temporal resolution of TEMPERA. We analysed temperature profiles only for cases with time-coincident radiosoundings (twice a day, at 11:00 and 23:00 UTC) to allow a direct comparison with radiosonde measurements. The radiosonde measurement with high vertical resolution were interpolated to the pressure grid of the TEMPERA retrieval. In total, 1319 all-weather

cases has been analysed. Cloud free situations are defined as cases where the ILW from HATPRO is $\leq 0.025\,\mathrm{mm}$ and the cloud cover from APCADA is 0 or 1 oktas. A case is defined as cloudy when the ILW is larger than $0.025\,\mathrm{mm}$ and the cloud cover from APCADA larger than 6 oktas. This ensures that only cases with relatively homogeneous cloud situations are considered as cloudy, whereas partially cloudy situations are excluded from the analysis. Rainy cases have been excluded for cloud free and cloudy cases based on a rain detector of the HATPRO instrument.



## 3 Methodology

This section presents the background information of temperature sensing by microwave radiometers and the retrieval settings for TEMPERA (Sect. 3.1). In addition, the different absorption species considered in the TEMPERA retrieval are presented in Sect. 3.2 with special focus on cloud liquid water (Sect. 3.3).

### 3.1 Temperature retrievals

The TEMPERA radiometer measures oxygen emissions to determine atmospheric temperature profiles. The volume mixing ratio of oxygen in the atmosphere is constant $21\%$ up to around $80\,\mathrm{km}$ and its concentration is only pressure dependent. Consequently, variations in measured intensity are due to variations in the molecular emissions and therefore in the temperature, and not due to differences in the oxygen concentration along the vertical profile. Applying this principle allows to obtain informa-

tion about the tropospheric temperature along the path of emitted radiation by comparing the intensity of different channels in the microwave spectrum. Even more information can be obtained when the instrument is measuring at different zenith angles, leading to measurements of different atmospheric paths. With this combination of spectral and angular information, one can infer the temperature distribution along the atmospheric profile from the measurements. This principle is based on the theory of radiative transfer. The relationship between the state of the atmosphere (temperature) and the measured signal (intensity)

is called *forward model*. In the TEMPERA retrieval, the forward model is given by the radiative transfer equation, where the measured signal is expressed by the brightness temperature $T_b$:

$$T_b(\nu, z_0) = T_0 e^{-\tau(z_1)} + \int\limits_{z_0}^{z_1} T(z) e^{-\tau(z)} k_a dz. \tag{1}$$

$T_0$ is the brightness temperature of the cosmic background radiation, $T(z)$ is the physical temperature at height $z$, $\tau$ is the optical depth, $z_0$ is the altitude at the surface, and $z_1$ is the altitude of the upper boundary of the atmosphere. The absorption in

the atmosphere by different species is represented by the absorption coefficient $k_a$ (see Sect. 3.2). In the TEMPERA retrieval, the radiative transfer calculations are carried out with the *Atmospheric Radiative Transfer Simulator 2* (ARTS2) that has been developed by Eriksson and Buehler (2011).

Since the measurements provide the intensity expressed by $T_b$, an inversion technique has to be applied to obtain the temperature in the atmosphere $T(z)$. Because the forward model is a complex function, its inversion cannot be determined in a

straightforward way. The difficulty is that the solution of the inverse is not unique, because the atmosphere is composed of many different constituents. Different possible combinations of those constituents can lead to the same radiative signal without knowing which of those possible combinations corresponds to the true state of the atmosphere. To solve this inverse problem, different possible solutions exist, all based on non-linear iterative schemes. The inversion technique applied in the TEMPERA retrieval is the *optimal estimation method* (OEM) according to Rodgers (2000). The software tool *Qpack2* is used (Eriksson

et al., 2005) that provides together with ARTS2 a complete retrieval environment. The package Qpack2 requires some a priori information to solve the inverse problem. The a priori temperature profiles as well as the a priori covariance matrices are acquired from monthly means of radiosonde measurements in Payerne from 1994 to 2011.



The altitude grid used in the retrieval has a resolution of approximately 100 m in the first kilometre, 300 m from 1-5 km, and 500 m from 5-10 km (Stähli et al., 2013). In this study, the vertical altitude grid of the retrieval is usually given in metres, even if the retrieval uses a pressure grid. To convert TEMPERA's pressure grid to altitude in meters we used the corresponding radiosonde measurements of pressure and altitude. For more information about the temperature retrieval of TEMPERA please

refer to Stähli et al. (2013).

### 3.2 Absorption species in the retrieval

In the forward model of the TEMPERA retrieval, the absorption and emission of important gaseous species have to be considered, namely oxygen ($O_2$), nitrogen ($N_2$), water vapour and liquid water. For each of those atmospheric constituents, the absorption and emission coefficients as well as the vertical profile have to be defined in the forward model. The absorption

species and their corresponding absorption model used in the TEMPERA retrieval are summarized in Table 2.

The vertical distribution of $O_2$ and $N_2$ is given by standard profiles for summer (May to September) and winter (October to April) for middle latitudes (FASCOD (Fast Atmospheric Signature CODE), Anderson et al. (1986)). For the vertical profile of tropospheric water vapour, we used an exponentially decreasing function, which is calculated with the measured water vapour density at the surface (volume mixing ratio from the weather station) and a scale height of 2000 m (Bleisch et al., 2011).

Considering cloud liquid water in the retrieval is much more complex than including the other absorption species, because the size and the amount of liquid water droplets is highly varying and hence no standard profile can be applied. In this study, liquid water profiles have been built based on cloud measurements, as later described in Sect. 3.3.

The influence of clouds on the TEMPERA retrieval is largest in the lower, more transparent channels (51.25-52.85 GHz) of the TEMPERA spectrum. This is because at these frequencies, the absorption coefficient of cloud liquid is similar to the

oxygen absorption coefficient (Navas-Guzmán et al., 2014; Stähli et al., 2013). This is the reason why Stähli et al. (2013) only used the 8 more opaque channels for retrievals where clouds were present. The present study aims to improve the use of all 12 channels to maintain the large amount of spectral information by considering cloud absorption in the retrieval. In the following analyses, the use of only the higher 8 channels (more opaque) in the retrieval is therefore always compared to the use of all the 12 channels.

Small water droplets mainly absorb microwave radiation, whereas larger rain drops and ice crystals scatter radiation. Scattering by hydrometeors is not considered in the retrieval of TEMPERA, because the Rayleigh scattering criterion ($2\pi r/\lambda \ll 1$) is generally met for microwave wavelengths, and scattering can therefore be neglected. However, for large hydrometeors such as rain drops, ice crystals or large cloud droplets, scattering can be important. Rainy cases have been excluded, and scattering on other hydrometeors is neglected in this study.

### 30  3.3 Cloud models in the retrieval

To study the effect of cloud liquid water on the TEMPERA retrieval, liquid water profiles (or cloud models) have been included based on the measured values of CBH and ILW. We included 9 different liquid water profiles into the TEMPERA retrieval to assess the influence of profile shape and liquid water amount. We used three different profile shapes (rectangular, triangular,





and logarithmic) with three different values of liquid water content (LWC) each. The first LWC value ($0.1\,\mathrm{g\,m^{-3}}$) is a low LWC that corresponds to a stratus or stratocumulus cloud with low liquid water amount (Brasseur et al., 1999; Salby, 1996; Heymsfield, 1993; Kneizys et al., 1996). The second value ($0.28\,\mathrm{g\,m^{-3}}$) is a typical value for a stratus layer (Hess et al., 1998; Kneizys et al., 1996; Brasseur et al., 1999), and the third value ($0.4\,\mathrm{g\,m^{-3}}$) represents a cumulus or nimbostratus cloud (Salby,

1996). An LWC of zero was assumed below the cloud base (CBH from ceilometer) and above the cloud top. The cloud top was determined by the cloud thickness, calculated with the measured value of ILW. For the rectangular profile (Fig. 2 (a)), the cloud thickness $dz$ was determined by assuming a constant LWC value within the cloud and calculated with $\Delta z$=ILW/LWC (Navas-Guzmán et al., 2014). The triangular profile (Fig. 2 (b)) has been calculated with a linear increase of LWC up to 2/3 of the cloud thickness ($\Delta z = 2\cdot$ILW/LWC) and a following linear decrease. The logarithmic profile is based on observations from

Korolev et al. (2007). For thin clouds (dz $\leq 500\,\mathrm{m}$) it is similar to the triangular profile but the LWC increases logarithmically with altitude up to $0.8\,dz$ (Fig. 2 (c)). For clouds with a thickness larger than $500\,\mathrm{m}$, the LWC in the logarithmic profile increases up to $0.3\,dz$, remains constant in the middle part of the cloud and decreases then linearly (Fig. 2 (d)).

## 4  Analysis of cloud data

In order to include appropriate cloud models in the temperature retrieval from TEMPERA, the aforementioned instruments have

been used to detect and characterize clouds. Because the presence and characteristics of clouds at the study site in Payerne are highly important for this study, we briefly analysed the available cloud data during the study period (2014–2015).

The frequency distribution of detected cloud base altitudes by the ceilometer in 2014 and 2015 are presented in Fig. 3. For each altitude range, the corresponding amount of ILW measured by HATPRO is indicated for low ILW (grey, ILW $< 0.8\,\mathrm{mm}$), medium ILW (green, $0.08\,\mathrm{mm} \leq$ ILW $< 0.12\,\mathrm{mm}$) and high ILW (blue, ILW $> 0.12\,\mathrm{mm}$). Rainy cases as detected by the rain

detector from HATPRO have been excluded from the analysis. We observed that most of the clouds in Payerne have a cloud base between 0 and $2\,\mathrm{km}$, with maximal amounts of clouds below $250\,\mathrm{m}$. Low clouds have a balanced amount of low and high liquid water. For higher clouds the amount of ILW is decreasing and above $4\,\mathrm{km}$ most of the observed clouds have a low amount of ILW. Three ranges of cloud base altitudes have been chosen for this study based on the observed frequencies of cloud base heights. Because of the high amount of clouds with low cloud bases ($< 250\,\mathrm{m}$) we have decided to define low

clouds as clouds with a CBH $< 500\,\mathrm{m}$. Medium clouds are defined for CBHs between $500\,\mathrm{m}$ and $2500\,\mathrm{m}$, which includes most of the observed clouds. The remaining cases (CBH $\geq 2500\,\mathrm{m}$) build the class of high clouds.

The ILW as measured by the HATPRO radiometer shows a frequency distribution that is almost exponential (Fig. 4). Most of the detected ILW cases (42.6 %) have an amount of total liquid water smaller than $0.01\,\mathrm{mm}$.

The cloud types detected by the sky camera show that most of the clouds over Payerne are stratocumulus clouds (27.4 %, Fig.

5). In 12.2 % of the cases, cirrus-cirrostratus clouds are present and 9.0 % of the cases are cirrocumulus-altocumulus clouds. In 27.0 % of the measurements the sky was detected as cloud free (clear). The remaining indicated types have a frequency smaller than 7 % each. The data furthermore show that cumulus clouds as well as cirriform clouds (cirrocumulus-altocumulus and cirrus-cirrustratus or cirrus mix) have a small amount of ILW. The rain clouds cumulonimbus-nimbostratus have a higher





amount of ILW than other types. For all the other cloud types however, which are mainly stratiform clouds, no dominant relationship between amount of ILW and cloud type can be established.

Two different approaches were available to detect cloud cover, namely the sky camera and the APCADA algorithm. The skewed least-squares line of the two data sets in Fig. 6 shows that for a covered sky (cloud cover larger than 5 oktas or

60 %), APCADA slightly underestimates the cloud cover compared to the sky camera, whereas the APCADA overestimates the cloudiness for small cloud coverage compared to the camera. However, this result has to be treated with caution because the relationship between cloud cover in oktas and percentage is nonlinear (WMO, 2008). Nevertheless, having a closer look at the cases where the camera has high coverage values whereas APCADA gives low values shows that many of them are cirriform clouds. This is consistent with Dürr and Philipona (2004) and Wacker et al. (2015) because APCADA is not able to detect high

cirrus clouds because of their low longwave downward radiation.

## 5   Analysis of tropospheric temperature profiles

To investigate the impact of clouds on the TEMPERA retrieval, temperature measurements of the years 2014 and 2015 were compared to radiosonde measurements. In a first step, a statistical analysis of all-weather cases and cloud free cases has been conducted (Sect. 5.1). Afterwards, non-precipitating cloudy cases have been analysed to determine the impact of clouds on

the TEMPERA retrieval. For those cloudy cases, a liquid water profile was included in the TEMPERA retrieval based on cloud measurements. First, a simple cloud model was used and the impact on the retrieval was investigated by comparing it to retrievals where no clouds were considered in the forward model (Sect. 5.2). After that, different liquid water profiles in shape and values have been tested in the retrieval to evaluate the sensitivity to the type of the cloud model (Sect. 5.3). Finally, to analyse the liquid water influence on the 8 frequency channels that are less sensitive to liquid water, cloud models have also

been included in retrievals that used only those 8 more opaque channels (Sect. 5.4).

### 5.1   Analysis of retrievals without cloud incorporation

To assess the quality of the TEMPERA retrievals when no clouds are included in the algorithm, we analysed all-weather and cloud free cases without considering clouds in the retrieval. Cases of all weather conditions (Fig. 7 (a)) show a good agreement between TEMPERA and the radiosondes for the retrievals that used the full spectral information (12f, blue line). The mean

bias (average of all temperature differences of TEMPERA and radiosonde) reaches a maximum value of 3.2 K for those cases $(1.7 \pm 3.5$ over all heights (Table 3)). When only 8 frequency channels (8f) are used in the retrieval, the mean bias is even below 1.7 K in the whole troposphere $(0.8 \pm 2.3$ over all heights (Table 3)). The bias is in general positive, indicating higher temperatures for TEMPERA than for the radiosonde measurements. The bias is low at around 500 m height and increases to 3.2 K at 3.2 km when all 12 channels are used (to 1.6 K at 2.5 km for 8 channels) and then decreases to almost zero at 10 km

height. These results indicate an overall good performance of TEMPERA with higher discrepancies to the radiosondes when all 12 channels are used and largest differences at around 3 km. The latter is probably due to horizontal drift of the radiosondes



and larger variability of the atmosphere at these altitudes than higher up. The high standard deviations above 2 km when 12 channels are used indicate that the variability of the bias between different cases is high.

The biases for cloud free cases are generally smaller than for all cases, with a bias below 2 K for 12 channels ($0.9 \pm 1.8$ over all heights) and even below 1 K for 8 channels in the whole troposphere ($0.3 \pm 1.8$ over all heights (Table 3)). Only close to the

surface the mean bias for clear cases is large (3 K), indicating an overestimation of the temperature by TEMPERA close to the ground. This feature is possibly due to small ground inversions mostly present in the evening cases and usually not detected by TEMPERA. The inclusion of a ground temperature measurement for the first grid point in the retrieval might be an appropriate measure to correct for this drawback (Navas-Guzmán et al., 2016).

Generally the discrepancies between TEMPERA and radiosonde measurements are largest when all 12 channels are used.

Initially, this was attributed to the influence of liquid water in the transparent channels, as it was already demonstrated by Navas-Guzmán et al. (2014). However, also cases without cloud liquid water (cloud free cases) do better agree with the radiosondes in the middle troposphere when only 8 channels are used, as displayed in Fig. 7 (b). This surprising result suggests that other reasons than the presence of liquid water in the retrieval may be responsible for the uncertainties that exist when all 12 channels are used.

## 15 5.2 Analysis of retrievals with a simple cloud model

In a first step of our analysis of cloudy cases, a simple rectangular liquid water profile has been included in the temperature retrieval using all 12 channels, based on the measured values of CBH and ILW. A constant LWC of $0.28\,\mathrm{gm}^{-3}$ was used, standing for a typical stratus layer. Based on the given LWC and ILW the cloud thickness was calculated (Sect. 3.3).

Figure 8 shows an example for a temperature retrieval during cloudy conditions where such a cloud model has been included

in the forward model. The sky camera indicates a stratocumulus layer (Fig. 8 (d)), and the relative humidity (RH) measurement from the radiosonde (Fig. 8 (c)) shows the approximate altitude and thickness of the cloud (where RH=100 %). The radiosonde humidity profile is only displayed in this case to present an indirect in situ measurement of the cloud. The humidity profiles however have not been used to built a liquid water profile for the TEMPERA retrieval to avoid a dependency on the 'rare' radiosonde launches (only twice a day). The liquid water profile that was used for the retrieval in this case, based on HATPRO

and ceilometer measurements, is given in Fig. 8 (b). By comparing it with the cloud indicated by the RH (Fig. 8 (c)), we observe a good agreement but a slightly higher cloud base detected by the ceilometer. The retrieved temperature profile of the presented case is shown in Fig. 8 (a). When no clouds are included, the retrievals using 12 or 8 channels (green and light blue lines) show differences of up to 5 K compared to the radiosonde measurement (red line). When the simple cloud model is considered in the retrieval, the radiosonde profile and the retrieved profile are in good agreement up to $\approx 9\,\mathrm{km}$ (dashed blue line). The

improvement in the middle troposphere compared to the retrieved profiles without considering clouds is thus substantial for this case.

Liquid water profiles have been included for all other cloudy situations in the same manner as described for the example case above. To evaluate the general effect of the cloud incorporation on the temperature retrieval, the mean bias of the profiles considering clouds in the retrieval are compared to the same profiles without considering clouds (Fig. 9).



When no clouds are considered, the mean bias when 12 (8) channels are used increases from below 1 K in the first kilometre to a maximum of 4.6 K at 3.6 km (2.2 K at 2.5 km) and decreases then again to 0.4 K (-0.7 K) at 10 km height (black and grey lines in Fig. 9). We observe high standard deviations of up to 5 K in the middle troposphere when all 12 channels are used (Table 3), indicating high variability between different temperature profiles. When the simple cloud model is considered in

the retrieval, we observe a marked improvement of the mean bias compared to the profiles without clouds. For both 8 and 12 channels, the mean bias remains below ± 1 K in the whole troposphere (up to 9 km) when clouds are included (blue and green lines). Only in the first 500 m of the mean profile the retrievals without the inclusion of LWC show slightly better results. The standard deviation of the retrievals with clouds is higher when 12 channels are used (4.1 K) compared to 8 channels (2.4 K), and slightly smaller when no clouds are considered (3.4 K (12f) and 2.3 K (8f), see Table 3). The latter indicates a higher

variability in the temperature profiles when liquid water is considered in the retrieval.

Contrary to our expectations based on the absorption coefficients of water for higher frequencies (Sect. 3.2), the profiles using only 8 channels are also affected when liquid water is included. However, the effect of the inclusion of liquid water on the retrieval is as expected much larger when all 12 channels are used.

Table 3 summarizes the mean biases and the standard deviations for the different investigated samples (Fig. 7 and Fig. 9) for

different altitude ranges. Averaged over all altitude levels, we observe the lowest mean bias for cloudy cases using all frequency channels and including a simple cloud model (0.1 K). This sample shows however also highest standard deviations (4.1 K). In summary we observe a large improvement compared to the profiles that do not consider a cloud model in the retrieval. This result suggests that the influence of liquid water on the TEMPERA retrieval is important and that the use of a simple cloud model in the forward model improves the retrieval substantially.

**5.3   Analysis of retrievals with different cloud models**

In a next step, we included 9 different cloud models into the TEMPERA retrieval to assess the influence of liquid water profiles having different shapes and LWC values.

**5.3.1   Results of different cloud models for all cloudy cases**

In total, we tested the 9 liquid water profiles in the retrieval of 293 cloudy cases, always using all 12 frequency channels. Some

outlier cases with high ILW values resulting in unrealistic thick clouds (ILW > 0.5 mm) or cases with a very high difference to the radiosonde (absolute mean error averaged over all altitudes > 10 K) have been excluded (18 cases in total). The mean biases, as well as the standard deviations, the root mean squared errors (RMSE) and the Pearson correlation coefficients have been calculated for retrievals using the different cloud models (Fig. 10). Fig. 10 (a) shows that the mean biases of the retrievals with different cloud models improve substantially between 1 and 8 km compared to retrievals without clouds, with values

between -1.3 and +1.2 K in the whole troposphere. They increase from around 0.5 K in the first kilometre to around 1 K at 2 km height, and decrease then to -1.3 K at 10 km height. Consequently, all 9 cloud models improve the retrievals compared to the retrievals without clouds by around 1 K (8f) or even up to approximately 3 K (12f) in the middle troposphere.





The different cloud retrievals have the same bias in the first kilometre of the profile, above this height the profiles differ by maximal 0.5 K. This indicates that using the distinct cloud models results on average only in small differences in the temperature profiles. For some individual cases however, the differences can be large, as proved by standard deviations of around 3 K at certain altitudes (10 (b)). The retrievals using rectangular liquid water profiles (0.28 or $0.4 \, \mathrm{g \, m^{-3}}$, red and orange) have slightly smaller mean biases in the middle troposphere, whereas the retrievals with a triangular profile of $0.1 \, \mathrm{g \, m^{-3}}$ (dark green) have a slightly higher bias than the retrievals with other cloud models.

The standard deviation increases with altitude from around 1 K near the surface to around 3 K at 9 km for most of the retrievals (Fig. 10 (b)). In the first kilometre, all retrievals have a similar standard deviation. Above 2 km however it is largest for profiles that included a rectangular cloud (0.28 and $0.4 \, \mathrm{g \, m^{-3}}$, red and orange lines) as well as for retrievals using all channels without a liquid water profile (black line), indicating higher variabilities for those profiles. The smallest standard deviation is found for the retrieval using only 8 channels without clouds (grey line). The RMSE and the correlation coefficient (10 (c) and (d)) show similar results with largest RMSE (smallest correlation) in the middle troposphere for retrievals without clouds (12f) and retrievals using the rectangular liquid water profile of 0.28 or $0.4 \, \mathrm{g \, m^{-3}}$. The linear relationship is in general strong for all profiles with correlations higher than 0.8 up to 9 km height (Fig. 10 (d)). The correlation decreases with altitude, being almost 1 in the first kilometre and 0.8 at 9 km. Above this altitude, it decreases sharply. Between 3 and 6 km, the profiles without any cloud model but using only 8 channels (grey line) have a slightly higher correlation coefficient than the other retrievals. In general we observe that the triangular and logarithmic profiles present slightly better results than the rectangular profiles with 0.28 or $0.4 \, \mathrm{g \, m^{-3}}$, with smaller standard deviations and higher correlation coefficients.

### 5.3.2 Results of different cloud models for different types of clouds

To determine whether specific liquid water profiles show better performances for certain cloud characteristics than others, we computed statistical means of cases with different cloud properties. For this we divided the sample in three classes of different cloud altitudes, according to the measured CBH (see Sect. 4). We computed the mean biases and correlation coefficients for these classes when different cloud models were considered (Fig. 11). In addition to the distinction of three CBH classes, we also classified the cases according to their amount of ILW as defined in Sect. 4 (Fig. 12).

For all of the six different cloud characteristics (low, medium and high CBH or ILW), including a cloud model improves the mean biases markedly compared to the retrieval using all 12 channels without including clouds. The improvement is largest for clouds with a large amount of liquid water (improves by about 5 K at an altitude of 4 km, Fig. 12 (c)), but also in all the other situations the improvement on the mean bias is clear. In most of the six presented cloud situations the mean biases also improve compared to retrievals that use only 8 channels without considering clouds. This is however not the case for low clouds (Fig. 11 (a)) and for clouds with a low amount of liquid water (Fig. 12 (a)), where the mean biases of retrievals with cloud models (coloured lines) and retrievals without clouds using 8 channels (grey lines) are similar. Furthermore, the retrievals that use 8 channels without clouds have often lower standard deviations (not shown) and sometimes higher correlation coefficients than the retrievals that include clouds in the forward model (e.g. Fig. 11 (d) and Fig. 12 (f)). This result suggests that including cloud models in the retrievals when 12 channels are used improves them, but does not always lead to better results than using 8





channels without considering clouds. Especially for low clouds the correlation coefficient in the middle troposphere is markedly higher for retrievals that use 8 channels without considering clouds (Fig. 11 (d)). For low clouds, using 8 channels without a cloud model might thus be a better option than using all channels with a cloud model.

The sensitivity to the chosen cloud model is largest for high clouds and for clouds with high ILW, as shown by a larger dispersion of mean biases and correlation coefficients than for the other cloud situations (Fig. 11 (c), (f) and Fig. 12 (c), (f)). For high clouds, the rectangular cloud models with $0.28$ or $0.4\,\mathrm{g\,m^{-3}}$ have a substantially smaller correlation coefficient, which suggests that these two cloud models are less appropriate to correct for liquid water in case of high clouds. The triangular cloud model with $0.1\,\mathrm{g\,m^{-3}}$ shows good results for high clouds (high correlation, Fig. 11 (f)), but worse results for low clouds (Fig. 11 (d)) or clouds with high ILW (Fig. 12 (f)). We therefore conclude that the logarithmic profiles as well as the triangular profiles with an LWC of $0.28$ or $0.4\,\mathrm{g\,m^{-3}}$ and the rectangular profile with $0.1\,\mathrm{g\,m^{-3}}$ are most suitable to correct for liquid water absorption, whereas the rectangular profiles with $0.28$ or $0.4\,\mathrm{g\,m^{-3}}$ are generally less appropriate.

### 5.4 Cloud models in retrievals with opaque channels only

As described in Sect. 5.2, we observed to our surprise an improvement of the retrievals with only 8 channels when liquid water was included. Moreover, the improvement of retrievals using 12 channels with clouds was not always clear compared to retrievals that used only 8 channels. For these reasons we analysed in a last step the improvement of retrievals that use only 8 channels when the different cloud models are included in the forward model (Fig. 13). Firstly our results display that the dispersion between the retrievals using different cloud models is smaller when 8 channels and higher when 12 channels are used. This observation indicates that the type of the chosen liquid water profile is less important for the use of 8 channels, which can be explained by their lower sensitivity to liquid water. Secondly we observed that for all retrievals that use 8 channels with different cloud models, the mean biases improve by more than $1\,\mathrm{K}$ compared to the retrieval without clouds (Fig. 13 (a), solid lines). This improvement was surprising because of the lower sensitivity of those channels to liquid water. Furthermore, the mean biases between $1.5$ and $5\,\mathrm{km}$ are even slightly smaller than the biases of the retrievals that use all 12 channels (dashed lines). This is especially true for low clouds (not shown). For the standard deviation, the RMSE and the correlation we observe similar results with slightly better values in the middle troposphere for retrievals that used only 8 channels (Fig. 13 (b–d)).

Our results show that using all the 12 channels leads generally to higher standard deviations and lower correlation coefficients than omitting the 4 transparent channels by using only the 8 channels. We therefore hypothesize that the transparent channels are prone to some uncertainties. Similar observations have been reported by Löhnert and Maier (2012) and Navas-Guzmán et al. (2016). According to Navas-Guzmán et al., the uncertainties in the transparent channels may arise from spectroscopic effects that are not well considered in the forward model. Another explanation might be the spatial drift of the radiosondes. The transparent channels retrieve information from higher altitudes than the more opaque channels, and the radiosondes might measure different air masses at these altitudes due to horizontal drift. Löhnert and Maier (2012) explained the uncertainties for transparent channels by their larger path length due to lower opacities in the atmosphere. Over a larger path, temperatures are less homogeneous, especially for high zenith angles that retrieve profiles closer to the surface, leading to higher uncertainties for the transparent channels. They found the best results when the transparent channels were only used in zenith measurements,





where the atmospheric path is shorter. Further investigations would be necessary to assess an optimal use of the transparent channels in the TEMPERA retrieval. Despite the uncertainties added by using the transparent channels we could show however that retrievals that use the transparent channels are highly influenced by liquid water and their improvement by considering cloud models is more important than for the opaque channels. Reducing the uncertainties of the transparent channels to better

benefit of the cloud correction is however a demanding task of future work.

## 6    Cloud effect on stratospheric temperature profiles

TEMPERA does not only retrieve tropospheric temperature profiles, but is also able to retrieve stratospheric temperature. The focus of the present study is the cloud effect on tropospheric temperature profiles. However, the influence of clouds on stratospheric retrievals have briefly been investigated by the mean of an example case. The stratospheric retrieval uses the

two pressure-broadened oxygen emission lines at 52.5424 and 53.0669 GHz. For the stratospheric retrievals, a tropospheric correction is performed to remove the effect of tropospheric emission (Navas-Guzmán et al., 2015). This correction takes into account the tropospheric emission of different constituents such as oxygen, water vapour or liquid water and removes it from the measured spectrum. For this purpose, the measured brightness temperature at ground is corrected by the tropospheric transmittance, assuming an isothermal troposphere with a mean temperature $T_m$ (Ingold et al., 1998). The transmittance is

obtained from $T_m$ and from the measured brightness temperature at the wings of the emission lines. The mean temperature $T_m$ is calculated from ground temperature using a linear model based on radiosonde measurements (Navas-Guzmán et al., 2015). To assess the effect of a more developed correction for clouds in the forward model of the stratospheric retrieval, we included a rectangular cloud model for one example case in the same manner as done for the tropospheric retrievals before. In the chosen example, including a liquid water profile in the forward model has only a small effect on the resulting stratospheric temperature

profile (Fig. 14). The differences between the retrieved profile with and without clouds are on the order of $10^{-4}$ K. We therefore conclude that the cloud effect on stratospheric temperature retrievals is negligible and that the applied tropospheric correction is able to account for absorption by cloud liquid water in the troposphere.

## 7    Discussion and conclusions

Our goal was to evaluate the effect of cloud liquid water on tropospheric temperature retrievals from a microwave radiometer.

For this purpose, clouds have been characterized using different cloud data and liquid water profiles have been considered in the retrieval of the microwave radiometer TEMPERA. Our results showed that temperature profiles improve when clouds are considered in the TEMPERA retrieval and that the retrieval is sensitive to the different cloud models.

The analysis of two years of cloud data showed that almost one third of the clouds present at the study site Payerne are stratocumulus clouds. This is however prone to some uncertainties, because the success rate of the cloud classification algorithm

can decrease to 50 % when multiple cloud types are present at the same time (Wacker et al., 2015). Further we found that the



total amount of liquid water is generally small for cumulus and cirriform clouds and higher for cumulonimbus-nimbostratus clouds than for other cloud types. For stratiform clouds, no dominant tendency for the amount of total liquid water was detected.

The presence of these different cloud types at the study site is important for the TEMPERA retrieval, because the distribution of liquid water within a cloud varies for different cloud types. We showed that the amount of liquid water in the cloud model and its shape has an effect on the resulting temperature profiles. All the 9 tested cloud models resulted on average in an improved retrieval, with best results for logarithmic and triangular shapes. The sensitivity of the retrieval to the chosen cloud model was on average large for high clouds and clouds with a large amount of liquid water, whereas it was smaller for other cloud types. For high clouds and for clouds with high ILW, we also found that the rectangular profile using an LWC of 0.4 or 0.28 $\mathrm{g\,m^{-3}}$ was markedly less suitable to correct for cloud liquid water absorption than the other cloud models. This might be linked to the thickness of the modelled cloud, which is generally smaller for rectangular profiles. A thinner modelled cloud with large LWC leads to a "more concentrated" correction effect, and might result in an overcorrection for liquid water in some cases.

Our findings also emphasize the need to reduce the uncertainties of the more transparent microwave channels used in the retrieval. The standard deviations and thus the variability between different cases was higher when the transparent channels were used than for the use of only 8 opaque channels. Moreover the retrievals without the transparent channels displayed good results not only for cloudy cases, but also for cloud free cases. These results indicate that other features than cloud liquid water may introduce uncertainties in the transparent channels. We therefore hypothesize that the use of the more transparent channels may be an important source of uncertainty in the retrieval. To our surprise, profiles that use only the 8 more opaque channels are also positively affected when liquid water is considered in the retrieval. The choice of the cloud model however is less important for those retrievals, and the potential of improving the retrieval is smaller than for retrievals that use the additional more transparent channels. Finally we showed that the cloud effect on stratospheric retrievals is small and that a general tropospheric correction is sufficient for stratospheric retrievals also in the presence of clouds.

To conclude, the liquid water correction for tropospheric temperature retrievals has been successful, but the observed uncertainties in the transparent channels would need to be reduced. Using less measurement angles for the transparent channels as proposed by Löhnert and Maier (2012) might be an idea, but this would require further investigations. Navas-Guzmán et al. (2016) showed that measuring at low zenith angles shows better results for the brightness temperature of the transparent channels in clear sky cases, but for cloudy conditions the results are still unclear. Our findings demonstrate the potential of a liquid water correction in microwave temperature retrievals. We have shown that tropospheric temperature retrievals can be optimized by considering clouds and that they are sensitive to the chosen vertical distribution and amount of liquid water. The results emphasize the importance of liquid water absorption for microwave measurements and imply that considering liquid water may generally improve tropospheric microwave retrievals.





## 8 Data availability

The ceilometer, HATPRO, APCADA and radiosonde data were available from MeteoSwiss, while the sky camera data were provided by the Physical Meteorological Observatory Davos, World Radiation Centre (PMOD/WRC). The TEMPERA data are available on request from Leonie Bernet (leonie.bernet@iap.unibe.ch).

5 *Acknowledgements.* We thank MeteoSwiss, in particular Alexander Haefele for providing the data and Dominique Ruffieux, Ludovic Renaud, Philippe Overney and Jean-Marc Aellen for hosting our instrument and for the support on-site, as well as the PMOD/WRC, especially Christine Aebi for providing the sky camera data. This work has been funded through the GAW-project "Fundamental GAW-parameters measured by microwave radiometry" by MeteoSwiss and through the COST action TOPROF ES-1303, SBFI-Nr. C15.0030 as well as through the Swiss National Science Foundation under grant 200020-160048 and 200021-165516.





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





**Table 1.** Central frequencies and bandwidths of the 12 tropospheric channels, as well as of the Fast Fourier Transform (FFT) spectrometer used for the stratospheric retrieval (Stähli et al., 2013).

| Channel | Frequency [GHz] | Bandwidth [MHz] |
|---------|-----------------|-----------------|
| 1 | 51.25 | 250 |
| 2 | 51.75 | 250 |
| 3 | 52.25 | 250 |
| 4 | 52.85 | 250 |
| 5 | 53.35 | 250 |
| 6 | 53.85 | 250 |
| 7 | 54.40 | 250 |
| 8 | 54.90 | 250 |
| 9 | 55.40 | 250 |
| 10 | 56.00 | 1000 |
| 11 | 56.50 | 1000 |
| 12 | 57.00 | 1000 |
| FFT | 52.4-53.2 | 800 |





**Table 2.** Profiles and absorption models of the different absorption species that are included in the forward model. For the absorption models, the tag names used in the Atmospheric Radiative Transfer Simulator (ARTS2) as well as the corresponding sources are provided.

| Species | Profile | Absorption Model |
|---|---|---|
| $N_2$ | Standard profile summer/winter | N2-SelfContStandardType |
| | *(Anderson et al., 1986)* | *(Liebe et al., 1993)* |
| $O_2$ | Standard profile summer/winter | O2-PWR93 |
| | *(Anderson et al., 1986)* | *(Rosenkranz, 1993)* |
| Water vapour | Exponential profile from ground measurement | H2O-PWR98 |
| | | *(Rosenkranz, 1998)* |
| Liquid water | Different profiles based on measured cloud properties (CBH and ILW) | liquidcloud-MPM93 |
| | | *(Liebe et al., 1993)* |



**Table 3.** Mean bias (TEMPERA - Radiosonde) with standard deviation for different samples, using all 12 frequency channels (12f) and only the 8 more opaque channels (8f). The cloud model used for cloudy cases with LWC was a rectangular model with an LWC of $0.28\,\mathrm{gm}^{-3}$.

| | All weather situations (n=1319) | | Clear sky (n=371) | | Cloudy without LWC (n=311) | | Cloudy with LWC (n=311) | |
|---|---|---|---|---|---|---|---|---|
| | Bias (K) | | Bias (K) | | Bias (K) | | Bias (K) | |
| | 12f | 8f | 12f | 8f | 12f | 8f | 12f | 8f |
| Below 1 km | $0.6\pm1.4$ | $0.7\pm1.5$ | $0.7\pm1.2$ | $0.8\pm1.3$ | $0.5\pm1.2$ | $0.6\pm1.3$ | $0.5\pm1.3$ | $0.5\pm1.3$ |
| 1 km–3 km | $2.7\pm3.4$ | $1.5\pm2.3$ | $1.7\pm1.7$ | $0.7\pm1.5$ | $3.5\pm3.2$ | $2.0\pm2.3$ | $0.6\pm4.2$ | $0.7\pm2.5$ |
| 3 km–10 km | $1.9\pm5.0$ | $0.5\pm2.9$ | $0.6\pm2.2$ | $-0.2\pm2.2$ | $3.0\pm5.0$ | $0.9\pm3.0$ | $-0.4\pm5.9$ | $-0.2\pm3.1$ |
| Total (0–10 km) | $1.7\pm3.5$ | $0.8\pm2.3$ | $0.9\pm1.8$ | $0.3\pm1.8$ | $2.4\pm3.4$ | $1.1\pm2.3$ | $0.1\pm4.1$ | $0.2\pm2.4$ |





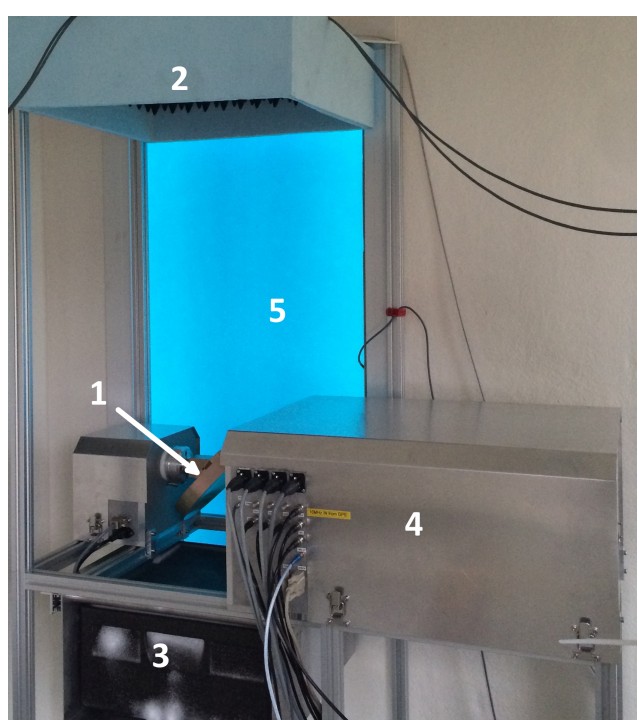

**Figure 1.** The TEMPERA (TEMPErature RAdiometer) instrument, with mirror (1), microwave absorbers (hot (2) and cold load (3)), receiver (4), and styrofoam window (5).





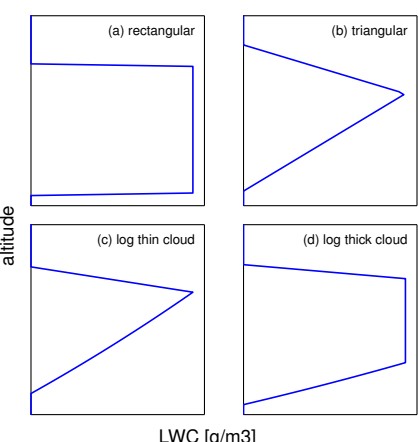

**Figure 2.** Schematic diagrams of different shapes of liquid water profiles that have been used in the temperature retrieval: (a) rectangular profile, (b) triangular profile and (c) and (d) logarithmic profile. A different logarithmic profile for thin (thickness smaller than 500 m, (c)) and thick clouds (thickness larger than 500 m, (d)) have been used.





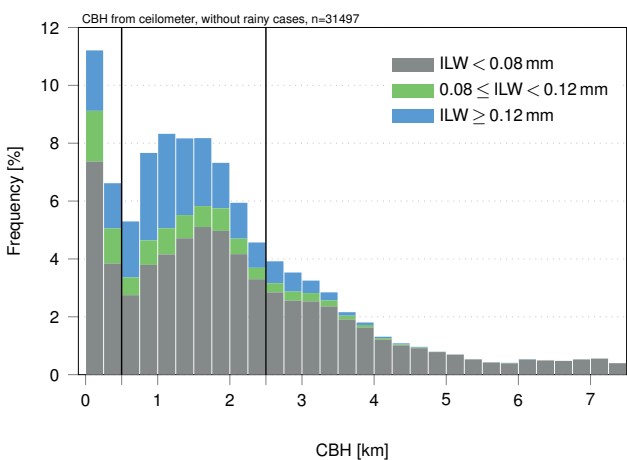

**Figure 3.** Cloud base heights (CBHs) from the ceilometer in Payerne with the corresponding integrated liquid water (ILW) from HATPRO (Humidity And Temperature PROfiler) in the years 2014 and 2015. The three different CBH classes (low, medium and high) are indicated by the vertical black lines.





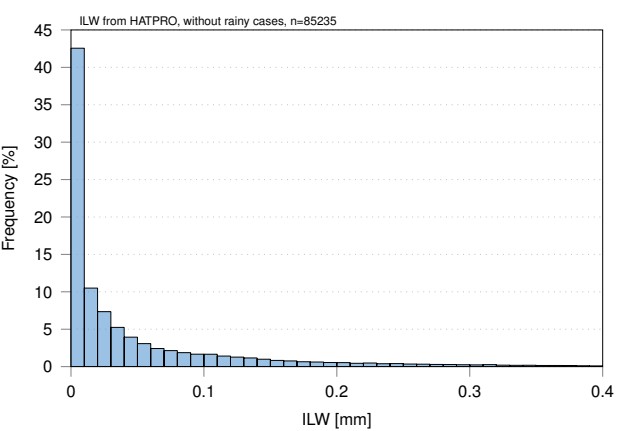

**Figure 4.** Integrated liquid water between 0 and 0.4 mm as retrieved by HATPRO in Payerne (2014–2015), without rainy cases.





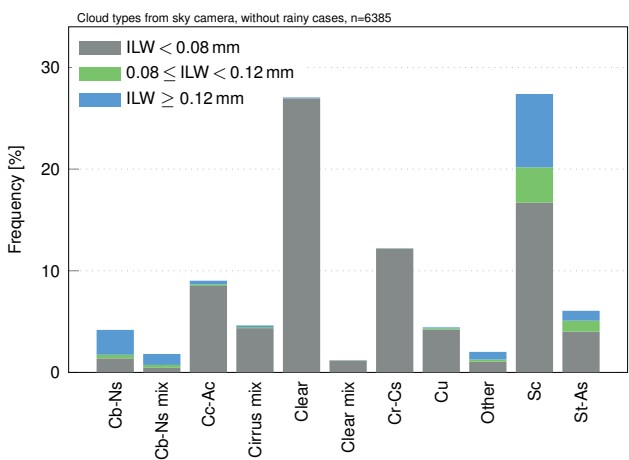

**Figure 5.** Cloud types detected by the sky camera in Payerne (2014–2015) and the corresponding ILW values from HATPRO. The different types are: cumulonimbus-nimbostratus (*Cb-Ns*), mix of cumulonimbus-nimbostratus and another type (*Cb-Ns mix*, containing Cb-Ns and Cu, Sc, or St-As), cirrocumulus-altocumulus (*Cc-Ac*), *cirrus mix* (containing Cc-Ac or Cr-Cs clouds mixed with Cb-Ns, Cr-Cs, Cu, Sc, or St-As), *clear* when it was classified as cloud free, *clear mix* (mix of cloud free and another type), cirrus-cirrostratus (*Cr-Cs*), cumulus (*Cu*), *other* (containing cases where Sc was detected together with Cu or St-As), stratocumulus (*Sc*), and status-altostratus (*St-As*).



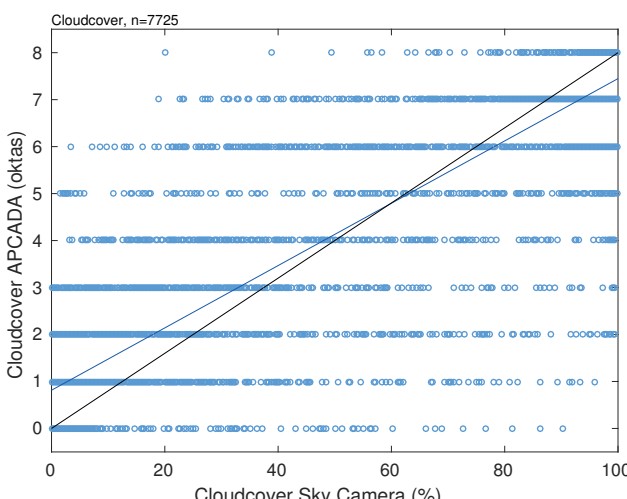

**Figure 6.** Cloud cover from APCADA (automatic partial cloud amount detection algorithm) vs. cloud cover from the sky camera for the years 2014 and 2015 with the least-squares fit (blue line).





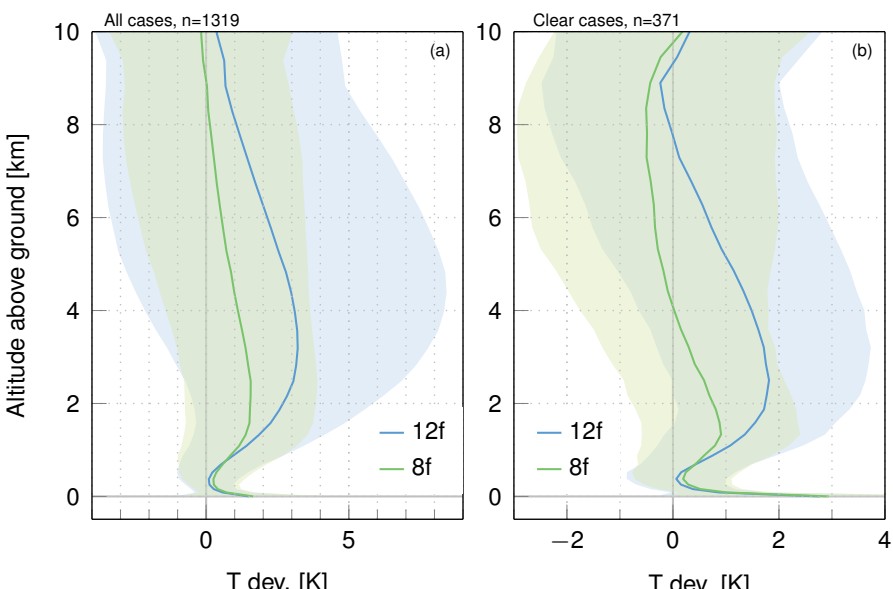

**Figure 7.** Mean bias (TEMPERA-radiosonde) with standard deviation (shaded area) using all 12 frequency channels (12f, blue) and only 8 channels (8f, green), for (a) all retrieved temperature profiles, and (b) cloud free cases in the years 2014 and 2015.





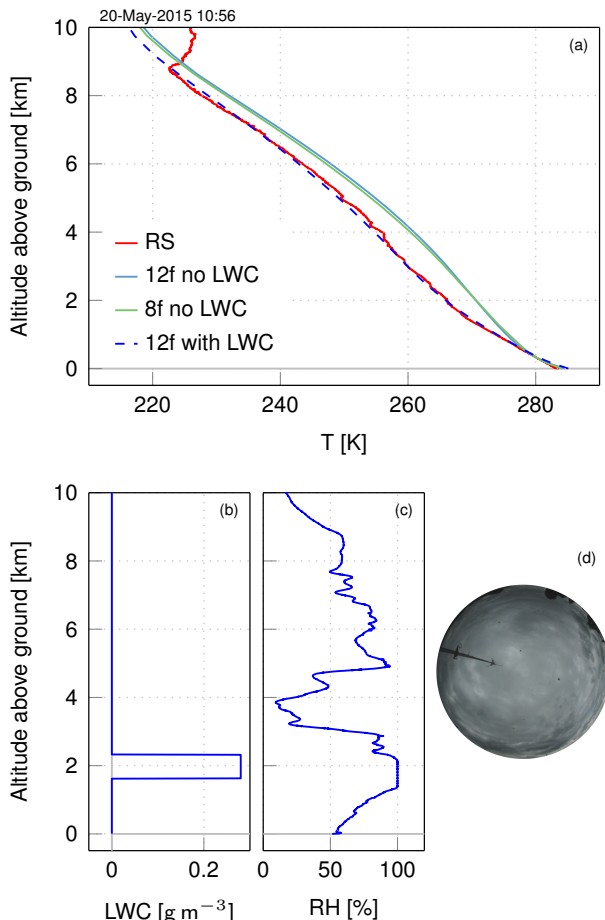

**Figure 8.** (a) Temperature profile retrieved by TEMPERA without considering clouds (light blue and green line) and with including a simple rectangular cloud model (dashed blue line). The corresponding radiosonde measurement is given by the red line. (b) Corresponding liquid water profile used in the retrieval with a maximal LWC value of $0.28\,\mathrm{gm^{-3}}$ (with a measured cloud base height of $1630\,\mathrm{m}$ above ground and an ILW of $0.192\,\mathrm{mm}$). (c) Corresponding relative humidity profile as measured by the radiosonde. (d) Corresponding sky camera image, indicating a totally covered sky with stratocumulus clouds. (Sky camera image from PMOD/WRC.)





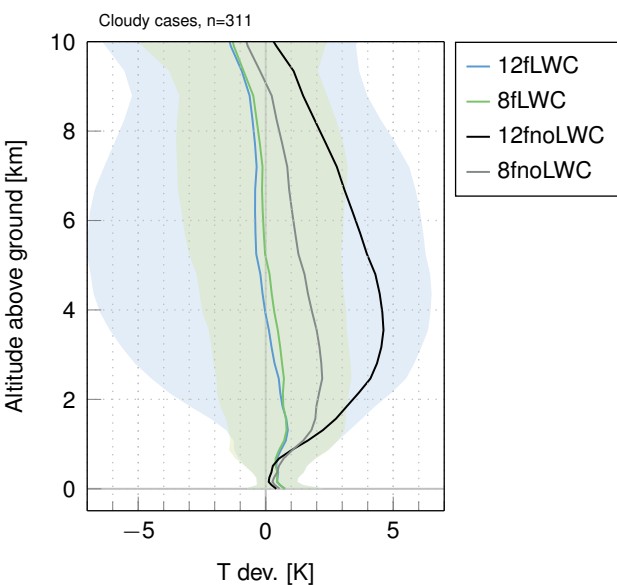

**Figure 9.** Mean bias with standard deviation (shaded area) when a simple cloud model is considered in the retrieval using all 12 frequency channels (12f, blue) and only 8 channels (8f, green) in the years 2014 and 2015. The mean biases for the same retrievals without considering clouds are given by the black (12 channels) and grey (8 channels) lines.





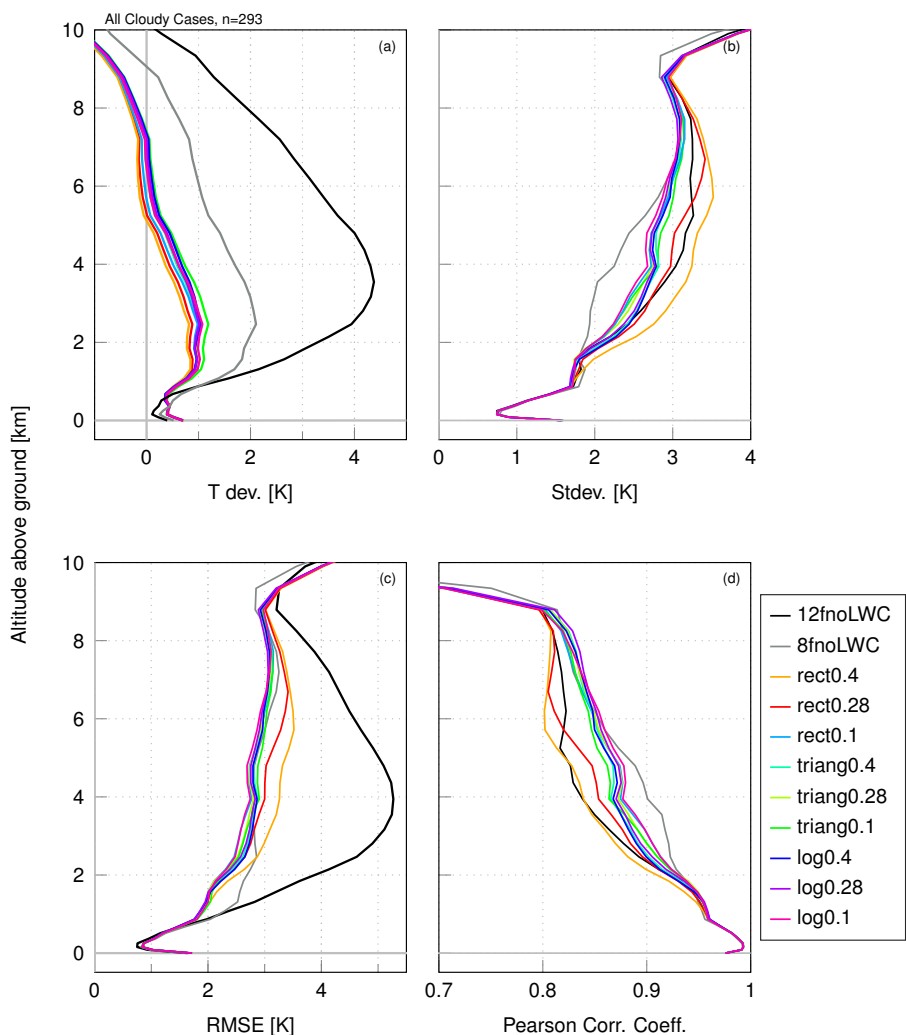

**Figure 10.** (a) Mean biases, (b) standard deviations, (c) root mean squared error (RMSE), and (d) correlation coefficients for retrievals during cloudy cases using different liquid water profiles (coloured lines). The values for retrievals without including cloud models are given by black (12 channels) and grey (8 channels) lines.




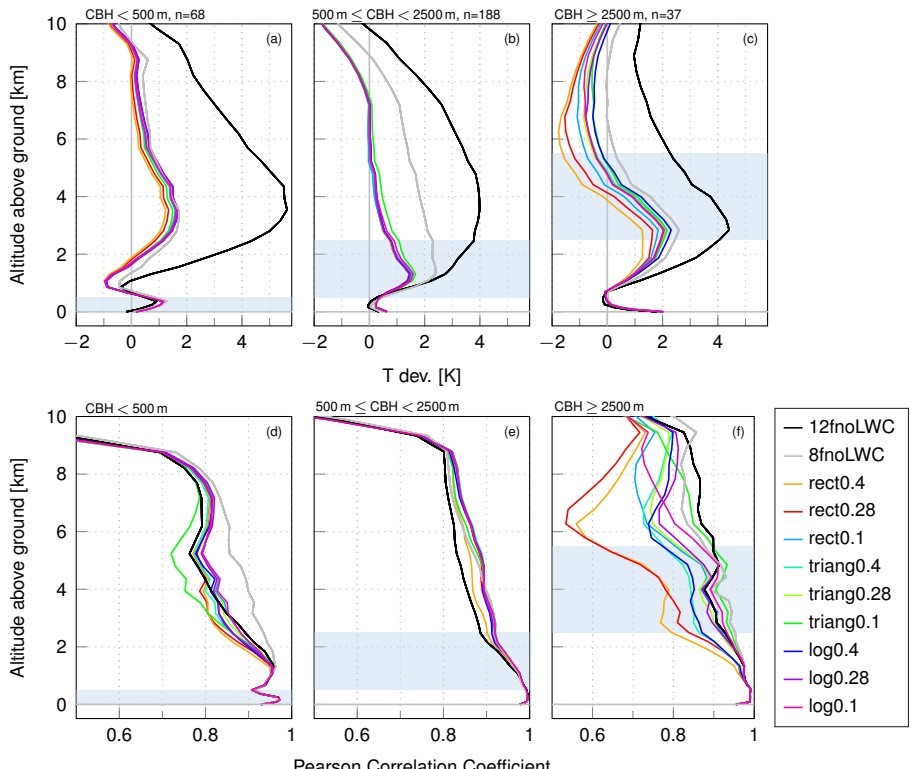

**Figure 11.** Mean biases (a–c) and correlation coefficients (d–e) for situations with different cloud altitudes. The blue shaded area illustrates the corresponding altitude range of the cloud bases. The different coloured lines represent retrievals that use all 12 frequency channels with different liquid water profiles in the forward model. The values for retrievals without including cloud models are given by black (12 channels) and grey (8 channels) lines.





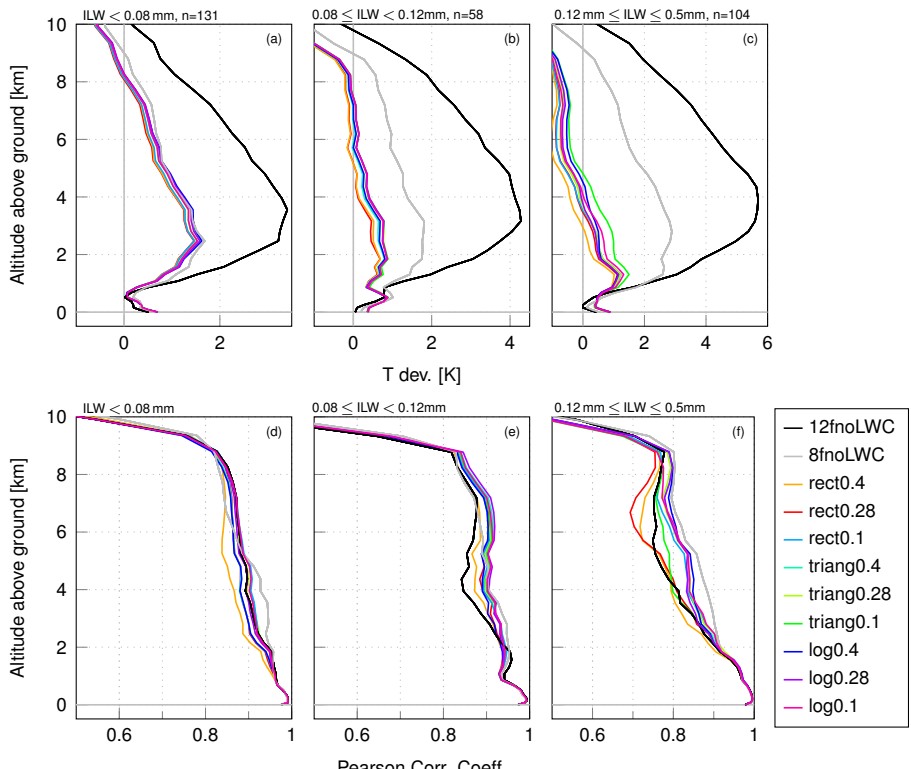

**Figure 12.** Mean biases (a–c) and correlation coefficients (d–e) for situations with different ILW. The different coloured lines represent retrievals that use all 12 frequency channels with different liquid water profiles in the forward model. The values for retrievals without including cloud models are given by black (12 channels) and grey (8 channels) lines.





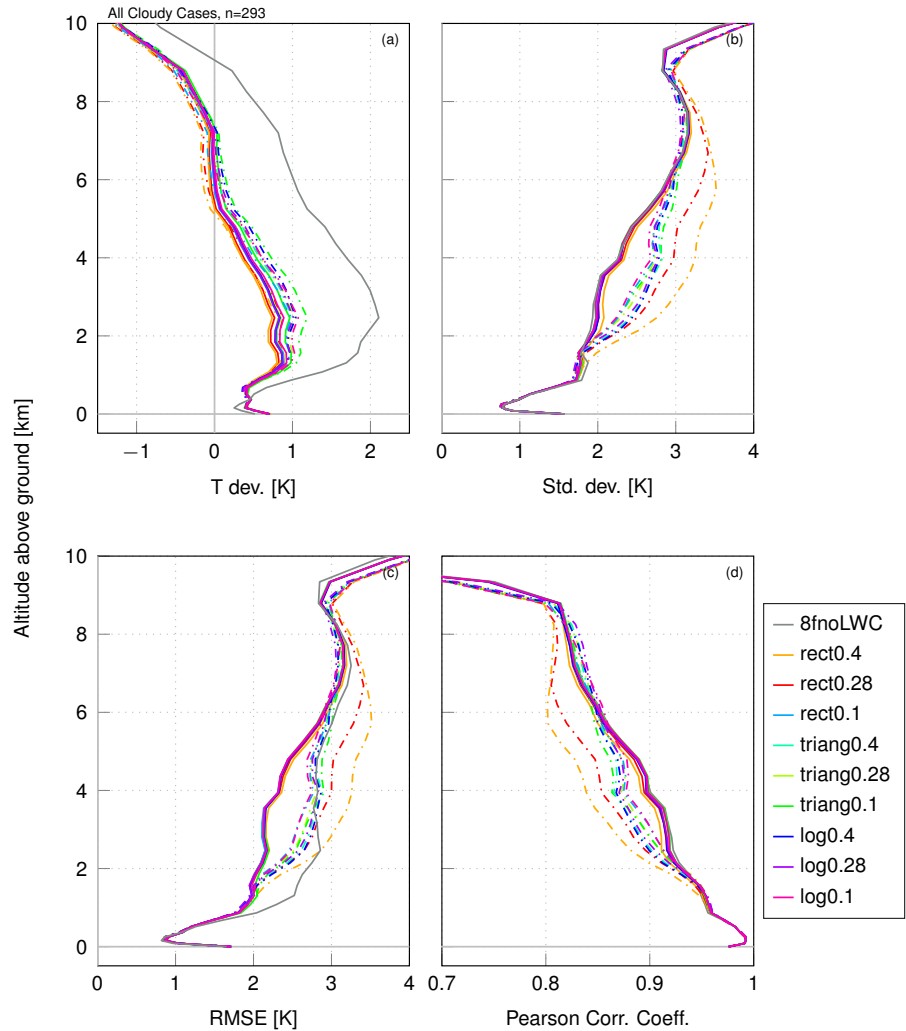

**Figure 13.** (a) Mean biases, (b) standard deviations, (c) RMSE, and (d) correlation coefficients for temperature retrievals during cloudy cases using different liquid water profiles. The solid lines represent retrievals that used only 8 frequency channels, retrievals that used all 12 channels (same as in Fig. 10) are represented by dashed lines. The retrievals using 8 channels without including clouds are represented by grey lines.



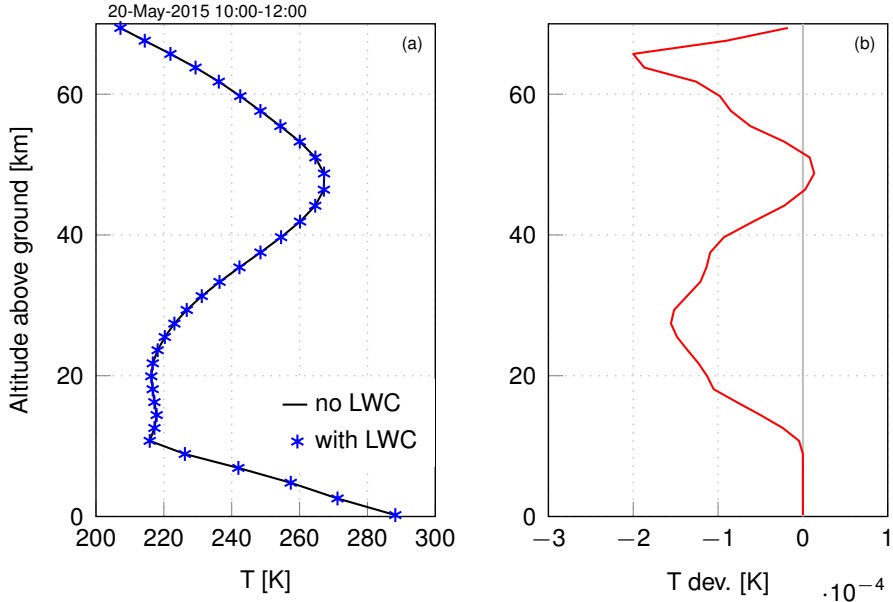

**Figure 14.** (a) Example case of a temperature profile in the stratosphere (same case as in Fig. 8) with and without clouds considered in the forward model. The included cloud model has a rectangular shape with a maximal LWC of $0.28\,\mathrm{g\,m^{-3}}$, a CBH at $1630\,\mathrm{m}$ and an ILW of $0.192\,\mathrm{mm}$. (b) Difference between the retrieved profiles with and without including clouds (retrievals without LWC - retrievals with LWC).