# Peer review of "The effect of cloud liquid water on tropospheric temperature retrievals from microwave measurements"

_Atmospheric Measurement Techniques, 2017_

## Referee Comment (RC1) · Anonymous Referee #1 · 5 Jul 2017

General Comments:

The paper is technically sound. I only have very minor comments listed below.

Specific comments:

Line 21: remove 'a'

Line 22: By traditional do you mean satellite. It is not clear to me.

Pg 6, line 26: change 'a' to 'the'

Pg 6, line 32: variance about the monthly mean profile at this site must be very large, especially in the cold seasons when there is more frontal activity. A-Priori will affect the retrieved profile of course. Has there been any study of this?
Pg 7, line 14: How sensitive is the retrieval to the exponential water vapor distribution

Pg 10, line 29: perfect???

Pg 11, line 32, : change 'a' to 'the'

―――――――――――――――――

---

## Referee Comment (RC2) · Anonymous Referee #3 · 13 Aug 2017

Comments to the manuscript: "The effect of cloud liquid water on tropospheric temperature retrievals from microwave measurements" by Bernet et al.

The authors analyze the effect of including cloud liquid water in the retrieval of temperature profiles and estimate the improvement to the profile caused by including a cloud model to the profile.

I must admit I am a little bit puzzled by the results and conclusions presented in section 5.3.2. Although I agree with the results that show that including LWP in the retrieval improves the temperature profiles (because it includes frequencies that are sensitive to the presence of liquid water) the results relative to the sensitivity of the retrieval to the liquid water distribution in the clouds contradict previous studies. For example, Crewell et al. (Geophys. Res. Lett., 36, 2009) showed that there is no sensitivity of microwave

brightness temperatures to the CLW profile. I am sure that the ILW retrievals used in the present paper are derived under that assumption. If there were such a sensitivity a LWC profile should be provided to retrieve ILW and IWV as well.

In my opinion to support the conclusions, the perturbation on the brightness temperatures caused by the various liquid profiles should be presented and it should be assessed whether the magnitude of such perturbation on the brightness temperature is large enough to justify the effect on the temperature profile retrieval.

---

## Author Comment (AC1) · 26 Sep 2017

Dear Referee 1,

thank you for having revised our manuscript and for your helpful comments. We are very grateful for your careful reading of the manuscript. Attached I send you the detailed response to your comments, as well as a marked-up version of the manuscript (uploaded as a supplement).

Kind regards,

Leonie Bernet

Please also note the supplement to this comment:

[Figure]

https://www.atmos-meas-tech-discuss.net/amt-2017-153/amt-2017-153-AC1-supplement.pdf

[Figure]

**Supplement:**

**The effect of cloud liquid water on tropospheric temperature retrievals from microwave measurements**

**Author's final response**

Leonie Bernet, Francisco Navas-Guzmán, and Niklaus Kämpfer

26th September 2017

Dear Reviewers,

thank you for your constructive comments on our manuscript during the discussion phase. We have taken the comments and helpful suggestions into account and are now submitting a detailed final answer to the comments of reviewer 1 and reviewer 2 and a revised version of the manuscript. The reviewer's comments are given in italic writing, our answers are given in normal typeface.

**Responses to referee #1**

In response to your suggestions we have taken the comments into account as follows (point to point response):

1. **Comment from reviewer 1:** *pg 1, line 21: remove 'a'*
   **Author's response:** We agree and performed the changes.

2. **Comment from reviewer 1:** *pg 1, line 22: By traditional do you mean satellite. It is not clear to me.*
   **Author's response:** By *traditional* we mean the well-established radiosonde measurements. We agree that this term is ambiguous and removed it.

3. **Comment from reviewer 1:** *pg 6, line 26: change 'a' to 'the'*
   **Author's response:** We agree and performed the changes.

4. **Comment from reviewer 1:** *pg 6, line 32: variance about the monthly mean profile at this site must be very large, especially in the cold seasons when there is more frontal activity. A-Priori will affect the retrieved profile of course. Has there been any study of this?*
   **Author's response:** We agree that the temperature at the study site varies highly within a month, especially during winter. However the impact of the a priori profile on the retrieved profile is small when the measurement response is high. Navas-Guzmán et al. (2016) for example showed that the retrieved temperature profile shows differences smaller than 0.6% when a simple linear decreasing a priori profile is compared to a profile from climatology. Furthermore, the variance of the a priori profile is considered in the a priori covariance matrix.

5. **Comment from reviewer 1:** *pg 7, line 14: How sensitive is the retrieval to the exponential water vapor distribution?*

    **Author's response:** We have analysed the sensitivity of the retrieved temperature profiles to the a priori water vapour profile by considering different artificial water vapour profiles (Fig. 1). An unrealistic profile such as a constant value in the whole troposphere (dashed cyan line) leads to a difference in the resulting temperature profiles of around 3K. However, the 2 tested more realistic profiles (black and green lines) lead to differences in the temperature profiles smaller than 0.5K.

    We therefore consider the used exponential approximation as an appropriate approximation. Bleisch (2010) has further shown that the exponential approximation does correspond very well to the IWV from GPS data.

6. **Comment from reviewer 1:** *pg 10, line 29: perfect???*

    **Author's response:** We replaced it by 'good agreement'.

7. **Comment from reviewer 1:** *pg 11, line 32, : change 'a' to 'the'*

    **Author's response:** We agree and performed the changes.

[Figure]

Figure 1: (a) Temperature profile retrieved by TEMPERA for different water vapour a priori profiles. 'exp' is the profile described in our manuscript, 'exp-0.001' is the same profile but with a lower ground value, 'const' is a constant profile in the whole troposphere, 'linear' is a linearly decreasing profile with pressure. The corresponding radiosonde measurement is given by the red line. (b) Difference between the different retrieved profiles and temperature profile that used the exponential approximation. (c) Different a priori water vapour profiles used in this example (see (a)).

**Responses to referee #3**

**Comment from reviewer 3:** *The authors analyze the effect of including cloud liquid water in the retrieval of temperature profiles and estimate the improvement to the profile caused by including a cloud model to the profile.*

*I must admit I am a little bit puzzled by the results and conclusions presented in section 5.3.2. Although I agree with the results that show that including LWP in the retrieval improves the temperature profiles (because it includes frequencies that are sensitive to the presence of liquid water) the results relative to the sensitivity of the retrieval to the liquid water distribution in the clouds contradict previous studies. For example, Crewell et al. (Geophys. Res. Lett., 36, 2009) showed that there is no sensitivity of microwave brightness temperatures to the CLW profile. I am sure that the ILW retrievals used in the present paper are derived under that assumption. If there were such a sensitivity a LWC profile should be provided to retrieve ILW and IWV as well.*

*In my opinion to support the conclusions, the perturbation on the brightness temperatures caused by the various liquid profiles should be presented and it should be assessed whether the magnitude of such perturbation on the brightness temperature is large enough to justify the effect on the temperature profile retrieval.*

**Author's response:** We thank the reviewer for this profound comment. Following the referee's proposition, we analysed the changes of the brightness temperature to different cloud models for a specific case. Figure 2 shows the effect of different cloud models on the fitted brightness temperature (Tb) for the same example case that was presented in Fig. 8 of the manuscript. The perturbation of Tb was analysed using three different cloud models (rectangular, triangular, and logarithmic) with a maximal liquid water content of 0.28g/m3 and a constant amount of integrated liquid water (Fig. 2 (b)). This analysis shows that some liquid water profiles result in a better agreement between fitted and measured Tb than others (Fig. 2 (a)). Futhermore, using different cloud models results in Tb differences of up to 7K at lower frequencies. For higher frequencies, the effect is negligible. This sensitivity in the lower frequency channels confirms the sensitivity of the retrieved temperature profiles to different cloud models.

These large differences for the use of different cloud models contrast with the results reported by Crewell et al. (2009). They observed differences smaller than 0.5K when comparing simulated Tb using an adiabatic or a constant liquid water profile. However, when comparing our study with the results of Crewell et al. (2009), we have to consider that they only used zenith observations, whereas our instrument measures at 9 different off-zenith angles. For zenith-observations, the total ILW in the observation path remains the same for the different cloud models. For off-zenith angles however, the total amount of liquid water in the observation path changes depending on the observation angle and on the used LWC distribution. We assume that this can explain the sensitivity of brightness and retrieved temperature to the different liquid water profiles in our study.

Furthermore the reviewer mentioned that in case of sensitivity to liquid water distribution in the clouds, it would be necessary to consider the LWC distribution also for ILW and IWV retrievals. Our study concentrates on temperature retrievals using the frequency range between 51 and 57 GHz, and does not analyse such a sensitivity for other frequency ranges that are used for ILW or LWC retrievals.

[Figure]

Figure 2: (a) Forward modelled brightness temperature (Tb) of TEMPERA at two different zenith angles (30° and 70°) obtained by using three different cloud models (rect (blue), triang (red), log (green)). The measured Tb is also shown (black dots). (b) Used cloud models in the forward model.

**Additional changes**

The figure about cloud types from the sky camera (Figure 5) in the manuscript was adapted. To have a more meaningful figure, the clear-sky cases were removed from the cloud statistics of the sky camera.

Additionally, some minor corrections have been performed (see red and blue marked text in the appended manuscript).

[revised manuscript text omitted]

---

## Author Comment (AC2) · 26 Sep 2017

Dear Referee 3,

thank you for having revised our manuscript and for your constructive comments. We are very grateful for your careful reading of the manuscript. Attached I send you our response to your comments, as well as a marked-up version of the manuscript (uploaded as a supplement).

Kind regards,

Leonie Bernet

Please also note the supplement to this comment:

[Figure]

https://www.atmos-meas-tech-discuss.net/amt-2017-153/amt-2017-153-AC2-supplement.pdf